



# An operational SMOS soil freeze-thaw product

Kimmo Rautiainen[1], Manu Holmberg[1], Juval Cohen[1], Arnaud Mialon[2], Mike Schwank[3/4],
Juha Lemmetyinen[1], Antonio de la Fuente[5], and Yann Kerr[2]

[1]Finnish Meteorological Institute, Erik Palménin aukio 1, 00560 Helsinki, Finland
[2]GAMMA Remote Sensing Research and Consulting AG, Worbstr. 225, 3073 Gümligen, Switzerland
[3]Swiss Federal Institute WSL, Zürcherstrasse 111, 8903 Birmensdorf
[4]CESBIO, Université Toulouse 3, CNES/CNRS/IRD/INRAe/UPS, 18 Avenue Edouard Belin, 31401 Toulouse cedex 9, France
[5]ESA - ESRIN, Largo Galileo Galilei 1, 00044, Frascati, Italy

**Correspondence:** Kimmo Rautiainen (kimmo.rautiainen@fmi.fi)

**Abstract.** The Soil Moisture and Ocean Salinity (SMOS) satellite is a valuable tool for monitoring global soil freeze-thaw dynamics, particularly in high-latitude environments where these processes are important for understanding ecosystem and carbon cycle dynamics. This paper introduces the updated SMOS Level-3 (L3) Soil Freeze-Thaw (FT) product and details its threshold-based classification algorithm, which utilizes L band passive microwave measurements to detect soil freeze-thaw
5    transitions; this is possible due to the difference in dielectric properties between frozen and thawed soils at this frequency band. The algorithm applies gridded brightness temperature data from the SMOS satellite, augmented with ancillary datasets of air temperature and snow cover, to generate global estimates of freeze-thaw state. A recent update to the algorithm includes improved noise reduction through temporal filtering. Validation results against *in-situ* soil moisture and temperature measurements and comparisons to ERA5 Land reanalysis data demonstrate the ability of the product to detect the day of first freezing,
10   an important metric for better understanding greenhouse gas fluxes and ecosystem dynamics, with improved accuracy. However, limitations remain, particularly in regions affected by radio frequency interference (RFI) and during spring melt periods when wet snow hinders soil thaw detection. Despite these challenges, the SMOS FT product provides crucial data for carbon cycle studies, particularly in relation to methane fluxes, as soil freezing affects methane emissions in high-latitude regions.

## 1   Introduction

More than half of the land in the Northern Hemisphere undergoes seasonal freezing and thawing each year, making it one of the most widespread environmental processes on Earth (Zhang et al., 2003). Seasonal soil freezing and thawing is not only a critical environmental phenomenon but also a key indicator of climate change and variability (Frauenfeld and Zhang, 2011; Peng et al., 2016). Soil freeze-thaw cycles are closely linked to surface temperature fluctuations and snow cover dynamics,
20   playing an important role in regulating the Earth's energy balance (Sokratov and Barry, 2002).



Monitoring the freeze-thaw cycle is essential because it directly impacts global ecosystems, hydrology, and climate systems. As soil freezes and thaws, it drives a range of ecological processes, including carbon and nutrient cycling, soil moisture dynamics, vegetation growth, and the activity of soil organisms. Thawing periods release stored water, influencing surface runoff, groundwater recharge, and the emission of greenhouse gases such as carbon dioxide and methane (Song et al., 2017; Wagner-Riddle et al., 2017; Boswell et al., 2020; Yang and Wang, 2019; Hayashi, 2013; Nikrad et al., 2016). These emissions are particularly relevant in the context of climate change, as thawing permafrost can release significant amounts of previously trapped carbon, creating a feedback loop that accelerates global warming (Johnston et al., 2014; Knoblauch et al., 2017). The freeze-thaw cycle also has substantial implications for infrastructure, as the freezing and thawing of soil can damage buildings, roads, and pipelines due to frost heave and ground subsidence. Agriculture is similarly affected, with the timing and intensity of freeze-thaw events influencing soil fertility, crop viability, and water availability (Kreyling et al., 2008; Krogstad et al., 2022). Therefore, accurate monitoring and prediction of soil freeze-thaw cycles are crucial not only for understanding natural ecosystems but also for mitigating risks and optimizing land-use practices in affected regions.

Global monitoring of the soil freeze-thaw cycle is vital for advancing our understanding of ecosystem dynamics, refining climate models, and managing natural resources. High-latitude and high-altitude regions are particularly sensitive to freeze-thaw cycles, where even minor changes can disproportionately affect local environments and contribute to broader global changes (Shiklomanov, 2012). L band passive microwave remote sensing is particularly effective for detecting soil freeze-thaw transitions due to the high contrast in permittivity between liquid water and ice at L band frequencies (1-2 GHz) (Rautiainen et al., 2014). Compared to higher frequencies, L band allows for deeper penetration into the soil, enabling observations several centimetres beneath the surface. As measurement frequency increases, the proportion of the signal originating from the soil decreases, with higher frequency bands interacting more strongly with surface vegetation or snow cover in winter. These subsurface observations are critical, as the significant difference in the dielectric constant between frozen and thawed soil results in pronounced changes in soil emissivity that L band radiometers can effectively detect, ensuring high sensitivity to freeze-thaw dynamics.

Over the past decades, several global data products have been developed to monitor soil freeze-thaw cycles. These include the Freeze-Thaw Earth System Data Record (FT-ESDR) (Kim et al., 2017), the Soil Moisture and Ocean Salinity Level 3 Soil Freeze-Thaw Product (SMOS L3FT) (ESA, 2023; Rautiainen et al., 2016), and the Soil Moisture Active Passive Freeze-Thaw Product (SMAP FT) (Derksen et al., 2017). The FT-ESDR combines data from the The Advanced Microwave Scanning Radiometer (AMSR-E) on NASA's Aqua satellite and the SSMIS on the Defense Meteorological Satellite Program platforms, providing a long-term, consistent dataset for global monitoring of freeze-thaw cycles, particularly useful for analyzing inter-annual variability and long-term trends. However, the FT-ESDR relies on high-frequency (36.5 GHz) radiometer data, which is primarily sensing the freeze-thaw status at the very surface of the landscape, and is therefore more affected by the vegetation and snow cover. In contrast, the SMAP FT and SMOS L3FT products are based on low frequency passive L band brightness temperatures, which are more sensitive to thermal emission originating from the soil. Although the SMAP mission originally included an active L band radar, the radar instrument unfortunately failed shortly after the mission's launch.



55    This paper describes the updated SMOS L3FT algorithm and introduces the dataset to the community (ESA, 2023). The SMOS L3FT product has been publicly available since 2018. Developed by the Finnish Meteorological Institute in collaboration with GAMMA Remote Sensing, Switzerland, the product is accessible through the European Space Agency (ESA) SMOS and the Finnish Meteorological Institue (FMI) dissemination services. In November 2023, the SMOS L3FT product underwent a major processor update from version 2 to version 3, with all data reprocessed.

## 2    Data

### 2.1    Data used for the soil freeze and thaw detection

#### 2.1.1    SMOS brightness temperatures

The ESA SMOS mission (Kerr et al., 2010), launched in 2010, was the first satellite mission to provide continuous L band observations covering the whole globe. For the SMOS L3FT product, the primary input data are the CATDS (Centre Aval de Traitement des Données SMOS) level 3 brightness temperatures (L3TB) dataset, version 331 (Al Bitar et al., 2017; CATDS, 2022). The L3TB data are in the ground polarisation frame, horizontal (H) and vertical (V) linear polarisations, and are provided in the Equal-Area Scalable Earth 2 (EASE-2) grid (Brodzik et al., 2012) with a polar projection at 25 km × 25 km grid cell size. On each overpass, SMOS measures an incidence angle profile of the brightness temperature. In the L3TB data the profiles are averaged into incident angle bins with 5-degree intervals. Daily CATDS files include all swaths observed over the Northern Hemisphere. The variables used are the H and V polarized brightness temperatures, their standard deviations and radiometric accuracies, number of views, number of views suspected to be affected by RFI, observation acquisition times, and incidence angles relative to nadir. The SMOS L3FT algorithm uses only data from the incidence angle bin of 50°- 55°.

#### 2.1.2    Two metre air temperature

Daily air temperature data at 2 metres above ground level are provided by the European Centre for Medium-Range Weather Forecasts (ECMWF). The operational L3FT processor utilizes the Atmospheric Model High Resolution 10-day Forecast data from ECMWF's real-time forecast system. During reprocessing, the near real-time air temperature data are replaced with ERA5-Land surface layer data, which are available with a latency of up to one month (Muñoz Sabater et al., 2021; Muñoz Sabater, 2019). The most recent reprocessing was performed in October 2023, and all data after 10 October 2023 have been processed using ECMWF near real-time data. ECMWF provides these data on a grid with a spatial resolution of 0.1° × 0.1° (approximately 11.1 km × 11.1 km at the equator, and 11.1 km × 5.6 km at 60° latitude), offering daily temperature values at 6-hour intervals (0, 6, 12 and 18 hours). The SMOS L3FT processor calculates the daily mean from these ECMWF air temperatures. The data are reprojected to the EASE-2 grid and resampled to a spatial resolution of 25 km × 25 km, using the nearest neighbor interpolation method.



### 2.1.3 Snow extent

The SMOS L3FT algorithm uses the global snow extent data produced by the United States National Ice Center (USNIC) using the Interactive Multisensor Snow and Ice Mapping System (IMS) (U.S. National Ice Center, 2008, updated daily). These IMS Daily Northern Hemisphere Snow and Ice Analysis data, originally in 4 km resolution with Polar Stereographic projection, are reprojected to the EASE-2 grid at 25 km × 25 km resolution using the majority interpolation method.

## 2.2 Data used for the validation

### 2.2.1 Soil moisture and soil temperature

The soil moisture (SM) and soil temperature (ST) data are obtained from the International Soil Moisture Network (ISMN) (Dorigo et al., 2011, 2021). Data are available from over 70 networks worldwide, seven of which provide near real-time updates. Here, ISMN data from six different networks are used to validate the SMOS freeze-thaw product. We use only data from those stations that measure both SM and ST from the top surface layer, at depths of 5 cm and/or 10 cm. These networks include SNOTEL - Snow Telemetry Network (Leavesley et al., 2008), SCAN - Soil Climate Analysis Network (Schaefer et al., 2007), USCRN - The U.S. Climate Reference Network (Bell et al., 2013), RISMA - Real-Time *In-Situ* Soil Monitoring for Agriculture Network (Ojo et al., 2015), BNZLTER - Bonanza Creek, the Long Term Ecological Research Network, and FMI - Finnish Meteorological Institute soil moisture and soil temperature observations (Ikonen et al., 2016, 2018).

### 2.2.2 ERA5 Land reanalysis data

The ECMWF ERA5 Land global atmospheric reanalysis dataset provides a consistent and long-term record of meteorological parameters over land surfaces (Muñoz Sabater et al., 2021). We used air temperature at 2 metres, soil temperature in layer 1 (0-7 cm depth), and snow depth. The data, provided on a $0.1° × 0.1°$ latitude-longitude grid, are reprojected to the EASE-2 grid at 25 km × 25 km resolution, consistent with the SMOS L3FT data, using the nearest neighbor interpolation method.

### 2.2.3 Land cover

The ESA CCI Land Cover time series v2.0.7 (1992 - 2015) data (ESA, 2017) are used to define the land cover distribution on the EASE-2 grid. The land cover classes were aggregated from the original 23 classes into 6 classes: agriculture, forest, low vegetation, wetland, open water and other (permanent ice, barren, urban). This aggregated land cover information was then used during the validation process to determine whether the land cover class at each *in-situ* sensor location represented the larger EASE-2 grid cell.



## 3 SMOS freeze and thaw (FT) retrieval algorithm

### 3.1 Algorithm outline

The SMOS FT detection algorithm is based on the physical principle that L band brightness temperatures vary significantly between frozen and thawed soils due to the distinct differences in their dielectric properties. Thawed soil contains liquid water, which has a much higher dielectric constant ($\epsilon' \approx 90$) at L band than the ice in frozen soil ($\epsilon' \approx 3.2$) (Mätzler et al., 2006). This large dielectric contrast directly influences the soil's emissivity and, consequently, the brightness temperature detected by the satellite. In the frozen state, the strong decline of free liquid water causes the soil's emissivity to increase, leading to an increase in the brightness temperature, similar to the effect of drying soil. In contrast, moist thawed soils exhibit lower emissivity due to the presence of liquid water, resulting in lowered brightness temperature.

To detect the FT state of the soil, the algorithm computes the Normalized Polarization Ratio (NPR), which we denote by $\Upsilon$ and is defined as:

$$\Upsilon \overset{\text{def}}{=} \frac{T_{\mathrm{B}}^{\mathrm{V}} - T_{\mathrm{B}}^{\mathrm{H}}}{T_{\mathrm{B}}^{\mathrm{V}} + T_{\mathrm{B}}^{\mathrm{H}}}, \tag{1}$$

where $T_{\mathrm{B}}^{\mathrm{V}}$ and $T_{\mathrm{B}}^{\mathrm{H}}$ are the vertically and horizontally polarized brightness temperatures, respectively. The advantage of NPR is its relative insensitivity to physical temperature variations, allowing it to robustly capture changes in soil moisture and FT transitions without the need for explicit temperature correction.

The algorithm employs a threshold-based classification method, identifying the soil state by comparing each observed $\Upsilon$ to empirically established frozen and thawed soil references, denoted by $\Upsilon_{\mathrm{fr}}$ and $\Upsilon_{\mathrm{th}}$, respectively. The resulting soil state estimates are further regularized by air temperature re-analysis data. The algorithm workflow scheme is shown in Figure 1 and described in details in sections below. As mentioned earlier, the SMOS FT algorithm primarily relies on CATDS L3 brightness temperature data as its main input The ascending and descending orbits are processed separately, resulting in two L3FT estimates for the two orbits.

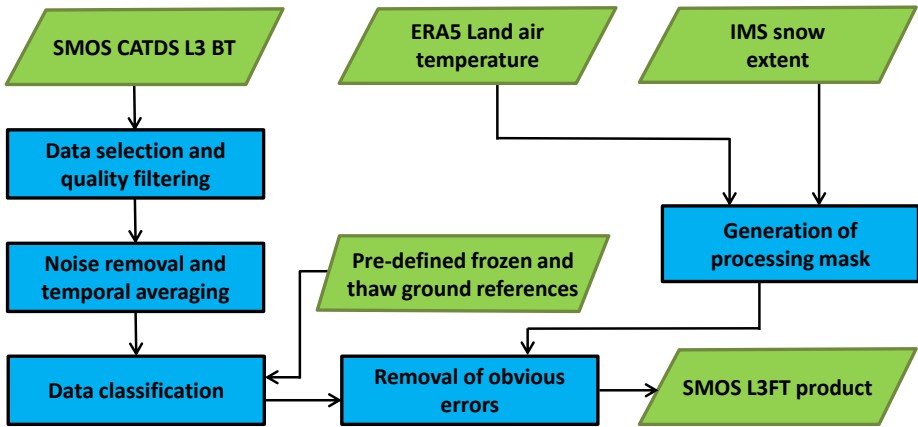

**Figure 1.** The SMOS soil FT detection algorithm workflow



## 3.2 Data selection and quality filtering

The brightness temperature measurements that are suspected to have reduced quality are filtered out. Table 1 summarises the quality filtering criteria. First, the brightness temperature values should be within the physically meaningful range. In the context of FT detection, values above 300 K can be omitted. Second, it is required that the incident angle bin contains at least 5 measurements. Third, the ratio

$$\chi = \frac{T_{\mathrm{B}} \text{ deviation}}{T_{\mathrm{B}} \text{ accuracy}} \tag{2}$$

between the sample standard deviation of the measurements and the average radiometric accuracy within the incident angle bin is expected to be bounded both from above and below with values 2 and 0.1, respectively. Fourth, the proportion of measurements suspected to be contaminated by RFI within the incident angle bin must be less than 40%.

**Table 1.** Data filtering criteria in the SMOS L3FT processor

| Description | Criteria |
| --- | --- |
| Realistic brightness temperature values | $0\,\mathrm{K} \leq T_{\mathrm{B}}^{\mathrm{V,H}} \leq 300\,\mathrm{K}$ |
| Sufficient amount of views within the incident angle bin | $N_{\mathrm{views}} \geq 5$ |
| Realistic sample deviation compared to radiometric accuracy | $0.1 \leq \chi \leq 2$ |
| Low RFI contamination | $N_{\mathrm{RFI}}/N_{\mathrm{views}} \leq 0.4$ |

## 3.3 Noise removal and temporal averaging

The individual SMOS L3 brightness temperatures, although averaged over the incident angle bin, contain noise that hinders the FT detection. To remove noise from the NPR time series computed from the L3TB, a temporal filtering is performed. In the SMOS L3FT processor, a simple Kalman filtering approach is used (Kalman, 1960; Särkkä, 2013). Every grid cell is filtered independently from each other, and the time series from a given grid cell is modelled as a dynamic linear model, so that

$$\begin{cases} \Upsilon(t_k) & = \Upsilon(t_{k-1}) + W_k, \\ \Upsilon_{\mathrm{L3TB}}(t_k) & = \Upsilon(t_k) + V_k, \end{cases} \tag{3}$$

where $\Upsilon(t_k)$ denotes the true physical NPR at time instance $t_k$, and $\Upsilon_{\mathrm{L3TB}}(t_k)$ denotes the noisy NPR that is computed from the L3 brightness temperatures at time $t_k$ by equation (1). $W_k$ and $V_k$ are the observation and model noise terms at time $t_k$, which were modelled as Gaussian random variables:

$$\begin{cases} W_k & \sim \mathcal{N}(0, w_k^2) \\ V_k & \sim \mathcal{N}(0, v_k^2). \end{cases} \tag{4}$$





The NPR observation noise variance $v_k^2$ and the process noise variance $w_k^2$ were estimated as follows:

$$
\begin{cases}
v_k^2 & = \frac{\operatorname{var} T_{\mathrm{B}}^{\mathrm{V}}(t_k) + \operatorname{var} T_{\mathrm{B}}^{\mathrm{H}}(t_k)}{\left(T_{\mathrm{B}}^{\mathrm{V}}(t_k) + T_{\mathrm{B}}^{\mathrm{H}}(t_k)\right)^2} \\
w_k^2 & = \vartheta^2(t_{k-1} - t_k),
\end{cases}
\tag{5}
$$

where $\vartheta$ is a tuning parameter, $T_{\mathrm{B}}(t_k)$ refer to the brightness temperature values at time $t_k$, and var$(\cdot)$ refers to the error variance of the brightness temperatures, which are provided in the data. The Kalman filter provides an optimal estimate of $\Upsilon(t_k)$ from the noisy time series $\Upsilon_{\mathrm{L3TB}}(t_k)$, balancing the noisy observations with their uncertainties to improve the signal

quality. Figure 2 shows an example of the observed time series before and after applying the Kalman filter. The advantage of the Kalman filtering approach over e.g., a running mean is that the observations are weighted according to their uncertainty, and in addition, the filtering parameter $\vartheta$ can be estimated from an observed time series by maximizing the likelihood of the observed time series with respect to $\vartheta$ (see e.g. book by Särkkä (2013)). The estimation is performed for the EASE-2 grid cell over Sodankylä, Finland, one of the applied validation sites (Ikonen et al., 2016), and the obtained value $\vartheta_0 = 0.003$ is used

globally.

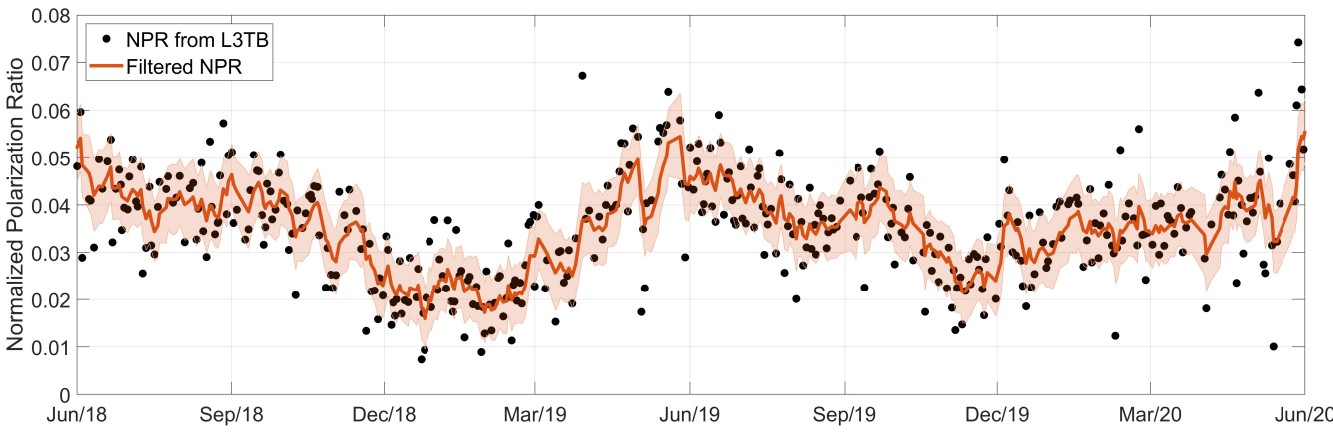

**Figure 2.** Time series of the non filtered (computed from the L3TB swath data) and the filtered normalized polarisation ratio from the EASE2.0 grid cell containing the Sodankylä validation site.

### 3.4 Frozen and thawed ground references

NPR varies between grid cells due to differences in land cover, soil properties, vegetation cover, and environmental conditions. As a result, each cell exhibits unique frozen and thaw soil references: $\Upsilon_{\mathrm{fr}}$ and $\Upsilon_{\mathrm{th}}$. To detect the freeze-thaw transitions, we scale the observed NPR signal:

$$
\Upsilon_{\mathrm{sc}} = \frac{\Upsilon_{\mathrm{th}} - \Upsilon}{\Upsilon_{\mathrm{th}} - \Upsilon_{\mathrm{fr}}},
\tag{6}
$$

where $\Upsilon_{\mathrm{sc}}$ is the scaled NPR. Note that $\Upsilon_{\mathrm{fr}}$ and $\Upsilon_{\mathrm{th}}$ are specific to each grid cell and they are empirically derived from the L3TB time series in conjunction with two auxiliary datasets: ERA5 Land air temperature and IMS snow extent. By scaling the





$\Upsilon$ values in this way, the algorithm adapts to the local conditions of each cell, enabling accurate determination of the soil state from the current observations.

The methodology used to define the reference values from the NPR time series is described below. If the daily mean air temperature was below -3°C and there was snow cover, the data were eligible for the frozen soil reference. Similarly, if the snow melt off occurred at least 28 days ago and the daily air temperature was above +3°C, the data were eligible for the thawed soil reference. This decision logic is shown in Figure 3. Reference values were derived from data collected between 1 January 2014 and 4 September 2023, with the end date limited by the availability of ECMWF ERA5 Land data at the time of

re-processing. The first years of data were excluded due to higher presence of RFI. From the selected period, all eligible frozen and thawed reference data were collected, and the 50 most extreme values were identified. The median of these values was used to define the frozen $\Upsilon_{fr}$ and thawed $\Upsilon_{th}$ reference values.

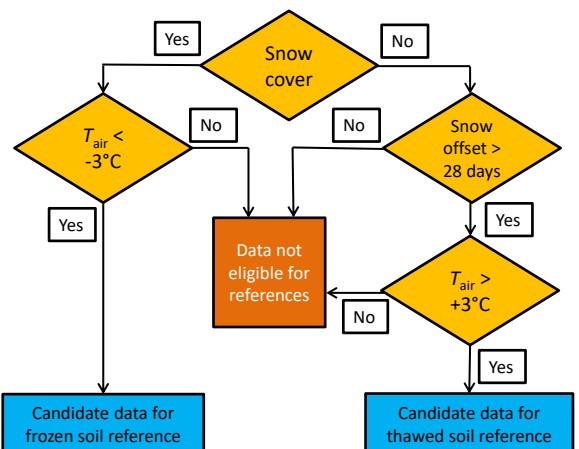

**Figure 3.** The logic for selecting the candidate data for the frozen and thawed soil references

## 3.5    Data classification

The FT class is estimated from the scaled NPR value $\Upsilon_{sc}$ according to table 2. The thresholds have been acquired in the

previous studies by fitting the scaled NPR value to frost tube observations in Finland (Rautiainen et al., 2016).

**Table 2.** Thresholds for the soil state categories in respect to parameter $A$ and in respect to frozen and thaw soil references.

| Category | Soil state | Condition |
|:---:|:---:|:---:|
| 1 | thaw | $\Upsilon_{sc} < 50\%$ |
| 2 | partially frozen | $50\% \leq \Upsilon_{sc} \leq 70\%$ |
| 3 | frozen | $70\% < \Upsilon_{sc}$ |





### 3.6 Removal of obvious errors and the processing mask

Even after the pre-processing steps for filtering the observational data, the initial freeze-thaw (FT) classification based on the scaled NPR value may contain errors, in particular over regions where some residual RFI is present, or where the separation of frozen and thawed references is small. Some of the obviously erroneous ground condition classifications can be mitigated using the auxiliary data: ECMWF air temperature and IMS snow extent. A processing mask (PM) was generated using these auxiliary data to estimate the season occurring in each grid cell. Additionally, the previously defined PM state restricted the selection of the new value. PM contains eight different values for four seasons (two for each).. They are described in Table 3 with the selection criteria and the allowed transitions.

**Table 3.** The nine values of processing mask $PM(t)$ for time $t$ (day), criteria for their conditions, the respective seasons, and allowed transitions $PM(t) \longrightarrow PM(t+1)$. The variables $T_{air}$ and $\overline{T_{air}}$ denote the daily mean and 10 day mean air temperatures, respectively.

| $PM(t)$ | Definition | Season | Definition criteria | Allowed transition $PM(t) \longrightarrow PM(t+1)$ |
|---|---|---|---|---|
| 0 | undetermined, initial value only | none | | 1, 3, 5, 7 |
| 1 | summer | summer | $T_{air} > 0°C$ or $\overline{T_{air}} > 0°C$ | 1, 2 |
| 2 | late summer | summer | $T_{air} \leq 0°C$ | 1, 2, 3 |
| 3 | freezing period, early phase | autumn | $\overline{T_{air}} \leq 0°C$ | 2, 3, 4 |
| 4 | freezing period, longer evolved | autumn | $\overline{T_{air}} \leq -1°C$ or $T_{air} < 0°C$ for 10 days | 3, 4, 5 |
| 5 | winter | winter | $\overline{T_{air}} \leq -3°C$ | 5, 6 |
| 6 | late winter | winter | $\overline{T_{air}} > 0°C$ | 5, 6, 7 |
| 7 | melting period | spring | $T_{air} > 3°C$ or $\overline{T_{air}} > 3°C$ | 5, 7, 8 |
| 8 | end phase of melting period | spring | $T_{air} > 3°C$ or $\overline{T_{air}} > 3°C$ and no snow | 1, 7, 8 |

PM affects the final estimate according to the following rules: (1) If $PM(t)$ is 3, 4, 7 or 8 (indicating freezing and melting periods), the mask has no effect. (2) If $PM(t)$ is 1 or 2 (indicating a summer period), all FT state estimates are forced into the thawed soil category. (3) During the winter period (when $PM(t)$ is 5 or 6), the mask prevents the soil state from changing towards the thawed state. However, the frozen state is not forced.

## 4 Validation

### 4.1 Validation with *in-situ* data

The soil freeze-thaw (FT) estimates were compared against the ISMN SM and ST data. The spatial and temporal differences between the satellite observations and the *in-situ* measurements create considerable challenges when interpreting the comparison results. The effective area of the observations differs significantly: *in-situ* sensors measure the soil parameters at a point





location, providing information from a very limited spatial area, while the effective footprint of the SMOS synthetic aperture radiometer observation varies from 30 to 50 km in size, depending on its location within the snapshot scene (McMullan et al.,
2008; Kerr et al., 2010). Temporally, *in-situ* data is continuous when the stations function nominally. The revisit time of SMOS varies with latitude: northernmost land areas are measured daily, while full global coverage is achieved twice every three days. However, in many regions, particularly in Eurasia, RFI can cause data gaps (Oliva et al., 2016), which significantly reduces the proportion of observations that can be used in the FT retrieval algorithm.

The SMOS FT product estimates the soil state at three levels (Table 2). To compare the SMOS FT estimates against the
*in-situ* data, a similar parameter indicating the soil state at the sensor location needs to be defined from the *in-situ* observations. The soil state at the *in-situ* sensor locations was quantified using a soil FT-index (SFTI). This index was derived by analyzing the relationship between the measured soil volumetric liquid water content (LWC) and soil temperature, represented by the soil freezing characteristic curve (SFCC). A simultaneous decrease in both LWC and temperature indicates soil freezing, while an increase in both parameters suggests soil thawing. This method is based on the approach developed by Pardo Lara et al. (2020)
and is further elaborated and explained in detail by Cohen et al. (2021). The SFTI is a site-specific metric representing the soil state, with values ranging from 0 to 1, where 0 corresponds to thawed soil and 1 to fully frozen soil. For comparison purposes, we used three SFTI thresholds: 50%, 70%, and 90%. The SFTI time series were then converted into three sets of binary data, each indicating whether the soil at the sensor locations was classified as either frozen or thawed based on these threshold values, with higher thresholds reflecting a stronger indication of frozen conditions. These binary datasets were compared with
the SMOS FT estimates. The day of first freezing (DoFF) in autumn was chosen as the comparison parameter because it plays a critical role in greenhouse gas (GHG) emissions, particularly methane. (Arndt et al., 2019; Tenkanen et al., 2021). Previous studies have shown that soil FT estimates derived from L band passive microwave data are most accurate during the autumn and cold winter periods. In the spring, direct observations from the ground, even at L band frequencies, are effectively blocked by the wet snow layer (Roy et al., 2015; Rautiainen et al., 2016). As a result, SMOS FT estimates during spring do not directly
reflect soil thawing but rather indicate the presence of a wet and melting snow layer. Consequently, *in-situ* sensors, which measure soil properties (SM and ST) directly, are not the most suitable ground reference for validating SMOS results during spring.

DoFF is defined here as the first day in autumn that is followed by at least 5 consecutive days of frozen soil. For SMOS data, an additional condition was applied: five consecutive observations must estimate a frozen soil state. Due to the varying
revisit time of SMOS, which depends on latitude, these five observations typically span 5–15 days. At high latitudes, SMOS provides daily coverage because the satellite's orbit intersects these regions more frequently. However, at lower latitudes, the revisit time increases, requiring a longer period to accumulate five observations from the same grid cell. This variability arises because the SMOS FT algorithm produces separate results for ascending and descending orbits, which further limits the frequency of available observations. Consequently, the DoFF derived from SMOS data may be delayed relative to the true
onset of soil freezing, reflecting the temporal limitations of satellite coverage. To quantify this uncertainty in the SMOS FT estimate, we identified the first time after which the soil state potentially changed to frozen, referred to as the day of first potential freezing (DoFPF). The period between DoFPF and DoFF represents the time when SMOS FT estimates indicate the





onset of soil freezing in autumn. Similarly, DoFF was determined from *in-situ* SFTI measurements using the three previously selected thresholds (50%, 70% and 90%) for comparison.

Figure 4 compares the day of first freezing (DoFF) derived from *in-situ* measurements with SMOS freeze-thaw (FT) product estimates, showing results for both ascending and descending orbits. The error bars indicate the range of uncertainty for both the SMOS FT product and the *in-situ* measurements in estimating DoFF. For SMOS, the error bars extend from the day of first potential freezing (DoFPF) to the day of first freezing (DoFF), with the midpoint marker representing the average estimate. The SMOS FT error bar reflects the variability in satellite observation times, which can span multiple days due to the satellite's
overpass frequency. For *in-situ* measurements, the error bars reflect the range between the 50% to 90% thresholds, with the marker also set at the midpoint. The error bars for *in-situ* data reflect the variability in defining the exact timing of freezing based on the SFTI thresholds. A wider range between the 50% and 90% thresholds suggests more gradual soil freezing, which introduces uncertainty into the determination of the DoFF. Narrower error bars suggest a more abrupt freezing transition, with the *in-situ* measurements providing clearer signals of soil state changes. The bias, Pearson correlation (R), and standard
deviation of difference (SDD) values were calculated for the midpoints. For SMOS, the result represents the effective FT state within the grid cell. For *in-situ*, the data may be from only one sensor location, or there may be several locations around the grid cell. If multiple sensors are included, the SFTI data were averaged considering the land class information of the sensor locations and the land class distribution of the associated grid cell. Prior to comparison, *in-situ* data were excluded if they were not representative of the larger EASE-2 grid cells. Several criteria for representativeness were given: (a) The land cover
similarity check with the aggregated land cover data (Section 2.2.3); the land cover on at least one sensor location had to be the same as the dominant land cover within the EASE-2 grid cell, the total land cover classes where the sensors were located had to cover 70% or more of the EASE-2 grid cell, and a maximum allowable fraction of 5% within a grid cell was permitted for open water, and likewise, the combined fraction of all types in the 'other' category (permanent ice, barren land, and urban areas) could not exceed 5%. (b) The Freezing Degree Days (FDD) check; for each EASE-2 grid cell and for each autumn/early
winter period, FDD were calculated using ERA5 Land air temperature data. If the FDD was $0°C$ or more than $500°C$ (i.e., the cumulative sum of daily freezing degree days) at the time when the *in-situ* sensor indicated frozen ground (at the 70% threshold), the *in-situ* sensor was considered unrepresentative of the entire grid cell area. (c) The soil frost depth (SFD) check; we estimated the expected average soil frost depth for each grid cell using a simple regression model based on ERA5 Land air temperature and snow depth data. The change in soil frost depth ($\Delta$SFD) was estimated using the regression model from
Gregow et al. (2011)

$$\Delta\mathrm{SFD} = a_1 + a_2 \cdot \mathrm{FDD}_{10} + a_3 \cdot d_{\mathrm{snow}}, \tag{7}$$

where snow depth $d_{\mathrm{snow}}$ is in units of centimeters, and the regression coefficients are $a_1 = 0.591\,\mathrm{cm}$, $a_2 = 0.079\,\mathrm{cm°C^{-1}}$, and $a_3 = -0.161$. $\mathrm{FDD}_{10}$ is the 10-day freezing degree days:

$$\mathrm{FDD}_{10} = \sum_{i=1}^{10} \max(0, -T_i), \tag{8}$$





and $T_i$ is the daily average temperature on day $i$ (in °C). Similarly, if the estimated frost depth was 0 cm or more than 100 cm when the *in-situ* sensor indicated soil freezing (70% threshold), the *in-situ* sensor could not represent the entire grid cell around it. As a result of the quality checks, the number of data points (N) in the comparison exercise was reduced from 550 to 131.

Tables 5 and 4 present the comparison metrics at various representativeness levels for SMOS DoFF and *in-situ* SFTI DoFF using the 50% threshold. The largest reduction in data points occurred during the land class similarity check (criterion a),
which also significantly improved the metrics. The FDD check (criterion b) identified nine additional cases where the *in-situ* soil freezing estimates clearly contradicted ERA5 Land data, resulting in noticeable improvements in the statistics. The final criterion (c), which involved comparison against model-based soil frost depth information, excluded 45 more cases and led to slight further improvements in the results.

**Table 4.** The comparison result metrics for ascending orbit SMOS DoFF and *in-situ* SFTI DoFF using the 50% threshold

| Similarity check | Number of data points (N) | Bias in days | Pearson correlation (R) | SDD in days |
|---|---|---|---|---|
| All data included | 550 | -14.6 | 0.33 | 34.1 |
| LC check (a) | 185 | -8.1 | 0.51 | 26.8 |
| FDD check (b) | 176 | -6.7 | 0.65 | 21.2 |
| SFD check (c) | 131 | -9.7 | 0.71 | 19.4 |

**Table 5.** The comparison result metrics for descending orbit SMOS DoFF and *in-situ* SFTI DoFF using the 50% threshold

| Similarity check | Number of data points (N) | Bias in days | Pearson correlation (R) | SDD in days |
|---|---|---|---|---|
| All data included | 550 | -15.8 | 0.31 | 34.0 |
| LC check (a) | 185 | -9.1 | 0.56 | 24.7 |
| FDD check (b) | 176 | -7.5 | 0.70 | 18.9 |
| SFD check (c) | 131 | -10.6 | 0.75 | 17.4 |

The metrics shown in Figure 4 demonstrate the performance of the SMOS FT product. For the descending orbits (Figure
4b), the bias is -6.3 days, with a Pearson correlation of 0.71 and a SDD of 18.6 days. This indicates that, on average, the SMOS product estimates the day of first freezing later than *in-situ* measurements. The relatively high correlation reflects a strong agreement between SMOS estimates and *in-situ* data, suggesting that the product reliably captures the freeze-thaw transition in autumn, despite the temporal and spatial differences between satellite and *in-situ* observations. The SDD highlights the deviation between the two datasets, which is typical considering the challenges of matching large scale satellite observations
to point-based *in-situ* sensors.

For the ascending orbits (Figure 4a), the bias is -5.0 days, with the same Pearson correlation of 0.71 and a SDD of 19.2 days. This suggests that ascending orbits tend to estimate freezing later than *in-situ* measurements, but 1.3 days earlier compared to estimates from the descending orbit. This earlier detection of freezing by ascending orbits aligns with the SMOS satellite's



sun-synchronous orbit configuration, where ascending orbits capture morning conditions (6 AM local time), and descending
orbits capture evening conditions (6 PM local time). The colder morning temperatures likely cause soil freeze-thaw transitions
to be detected slightly earlier during ascending passes.

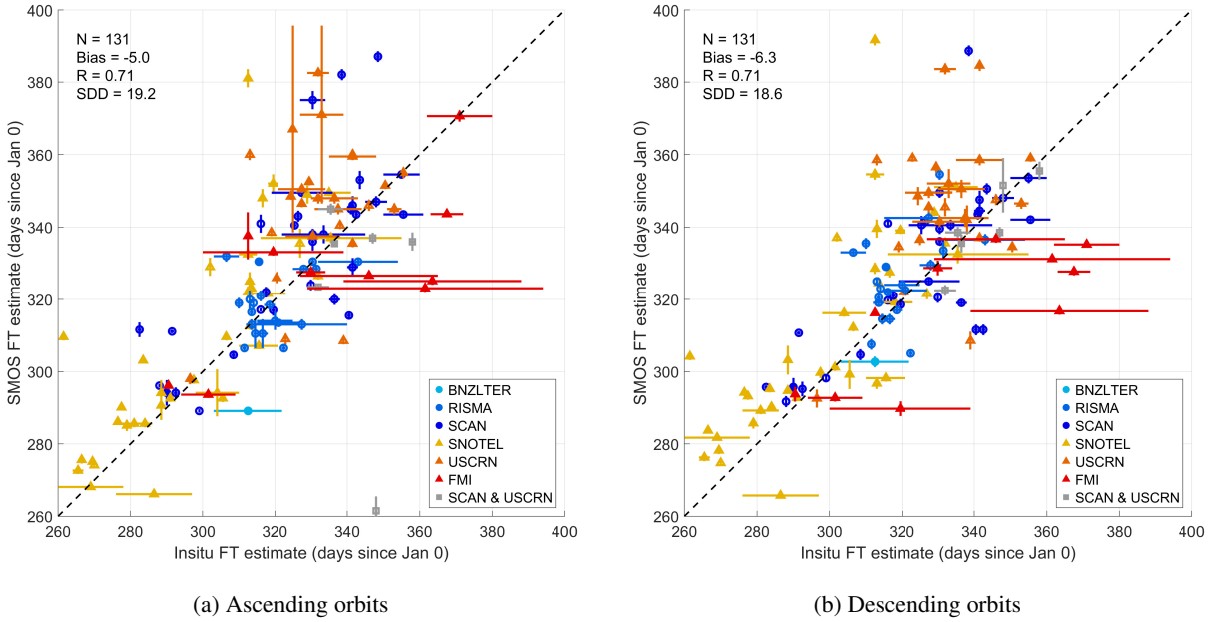

(a) Ascending orbits    (b) Descending orbits

**Figure 4.** Comparison of the day of the first freezing (DoFF) between SMOS L3FT and *in-situ* data. (a) Horizontal axis: DoFF from the
*in-situ* data with the error bar derived from thresholds 50% to 90%, marker set at the midpoint. Vertical axis: Estimates from the SMOS
ascending orbit data, the lower end of the error bar corresponds to DoFPF (day of first potential freezing) and the higher value corresponds
to DoFF, with the marker set to the centre. (b) Same as (a) but for descending orbit.

## 4.2 Comparison with ERA5 Land soil temperature data

We compared the SMOS FT with ERA5 Land soil temperature (level 1 representing depth 0-7 cm) product to analyse their
differences and compatibility. From the two products, SMOS is an observation-based product sensitive to the dielectric changes
associated with soil freezing, while the ERA5 is a model-based product representing the temperature of the soil. The two
products were compared by deriving a day of the first freezing (DoFF) from each data sets for each freezing period between
2010 and 2024.

The days of the first freezing were averaged over the freezing periods to have average DoFF. Figure 5 shows maps of the
average DoFF derived from SMOS FT ascending and descending orbits separately, and ERA5 soil temperature product. SMOS
FT and ERA5 show similar patterns of DoFF, particularly at the high latitudes. Differences are visible especially at the lower
latitudes where SMOS has less frequent observations. In addition, different latitudes are expected to show different DoFF
dynamics, and in Eurasia SMOS suffers significantly from RFI.



Figure 6 shows scatter plots comparing the mean days of the first freezing between the data sets. The associated statistics are shown in Table 6. In general, SMOS seems to estimate later freezing that the ERA5 soil temperature would indicate (13-14 days on global average). Possible reasons for this difference include: 1) the SMOS observation frequency; it is possible that SMOS observes the freezing later simply due to delayed good quality observation with respect to the soil freezing. RFI is a usual disruption to the SMOS observations. 2) The estimation of the day of the first freezing from the two data sets is slightly different, as the SMOS observation times have to be accounted for. 3) Systematic errors in the ERA5 soil temperature data. In particular the first freezing is derived by looking at the time when the soil temperature drops below 0°C. This estimate might be sensitive to errors in the modelled temperature values.

Furthermore, land cover distribution within the SMOS footprint affects the SMOS FT performance. High areal coverage of forest and water bodies on one hand dampen the observed FT signal, making the freeze-thaw detection more difficult, and on the other hand create their own contribution to the SMOS observation that are not fully accounted for in the SMOS FT product.

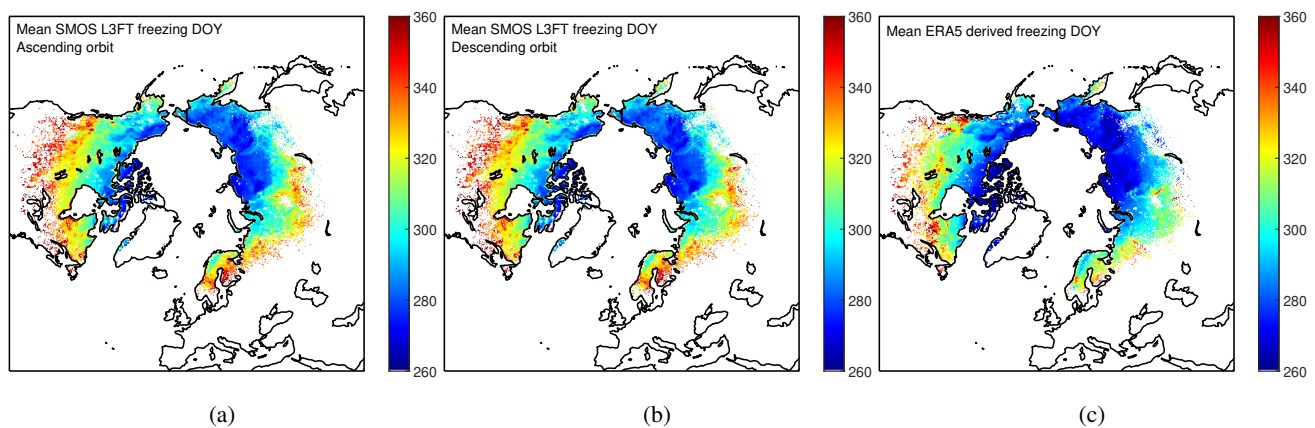

(a)  (b)  (c)

**Figure 5.** Average DoFF for the freezing periods between 2010 and 2024, a) SMOS FT product with ascending orbits, b) SMOS FT product with descending orbits, and c) ERA5 Land soil layer 1 temperature derived first freezing. Average values where the freezing day was successfully estimated from the data for 10 or more years are shown. The ERA5 derived mean freezing day is shown only for those values where also SMOS FT has a successful estimate.



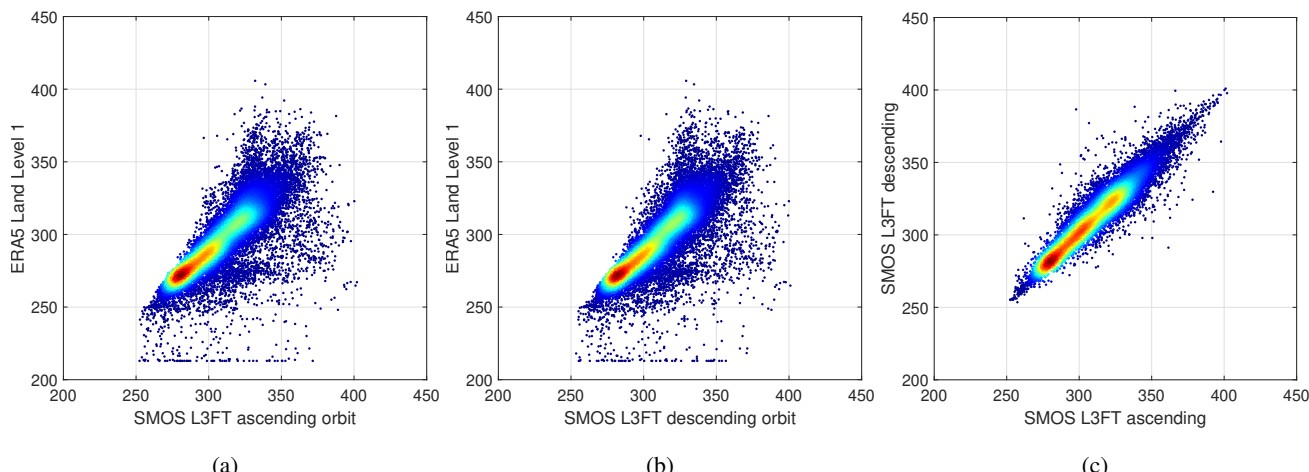

(a)           (b)           (c)

**Figure 6.** Scatter plots comparing the average DoFF between a) SMOS FT ascending orbit and ERA5 Land derived, b) SMOS FT descending orbit and ERA5 Land derived, and c) SMOS FT ascending and descending orbits.

**Table 6.** Statistics corresponding to Figure 6.

| Case | Bias (days) | SDD (days) | R |
|---|---|---|---|
| ERA-ASC | -13.0 | 14.2 | 0.81 |
| ERA-DSC | -14.3 | 14.2 | 0.81 |
| DSC-ASC | 1.4 | 5.9 | 0.97 |

# 5 Product limitations

## 5.1 General limitations

The SMOS FT retrieval algorithm detects permittivity changes caused by the phase transition (or change in the aggregate state) of liquid soil water to ice. Due to the basis of the method, areas with dry soils are challenging as the annual variability of soil permittivity due to soil freezing is minimal, resulting in minimal dynamics of the brightness temperature signal. Also, areas with a very thin or non-existent soil layer (e.g. rocky areas and mountains) are challenging. At L band, the typical penetration depth ranges from a few centimetres to 10-15 cm, depending on the amount of free liquid water in the soil. Therefore, the detection of soil conditions based on L band observations is limited to the near-surface layer, which is still significantly thicker compared to the surface layer detected by higher frequency radiometers and optical sensors.





## 5.2 Spatial and temporal coverage

SMOS observations cover the entire globe twice in three days. The northernmost land areas have daily overflights due to
the orbit configuration. Prior to the SMOS mission, passive L band microwave observations were only made in space during
the Skylab 3 mission in 1973 (T. J. Jackson and Eagleman, 2004). The revelation of strong presence of man-made RFI in
the protected frequency band (1400 - 1427 MHz) following the SMOS launch was a surprise (Oliva et al., 2012, 2016).
As a consequence of the RFI level, spatial coverage over the Eurasian continent is severely hampered, moreover increasing
significantly over Eastern Europe after 2022. Figure 7 shows the average observation interval (in days) of the SMOS FT product
for the period 1 June 2010 - 31 December 2021 for ascending and descending orbits. The more frequent observations towards
the north due to the orbit configuration is clearly visible. The presence of RFI increases towards the south on the Eurasian
continent and primarily affects the descending orbit observations due to the forward tilt of the instrument. The North American
continent is much less affected by RFI contamination, except for the first years of SMOS operations. Figure 8 shows the average
observation interval (in days) for the period 1 January 2022 to 1 June 2024. The increased RFI contamination over Europe is
clearly visible hampering the SMOS FT product over a considerable area.

An important feature of the SMOS FT product is that it contains data indicating the date of the last acquired observation
for each location. This information is crucial for interpreting the data accurately, as it allows users to assess the timeliness of
the observations. If the last observation was acquired several days before the release of the current product, there may be a
significant gap in the data. During this period, changes in the soil state, such as a transition from thawed to frozen conditions,
could have occurred at any point between the latest observation and the current product date. Users must be aware that large
gaps in observation frequency can introduce uncertainty in the soil state estimates, making it essential to consult the last
observation date when analyzing the product data.



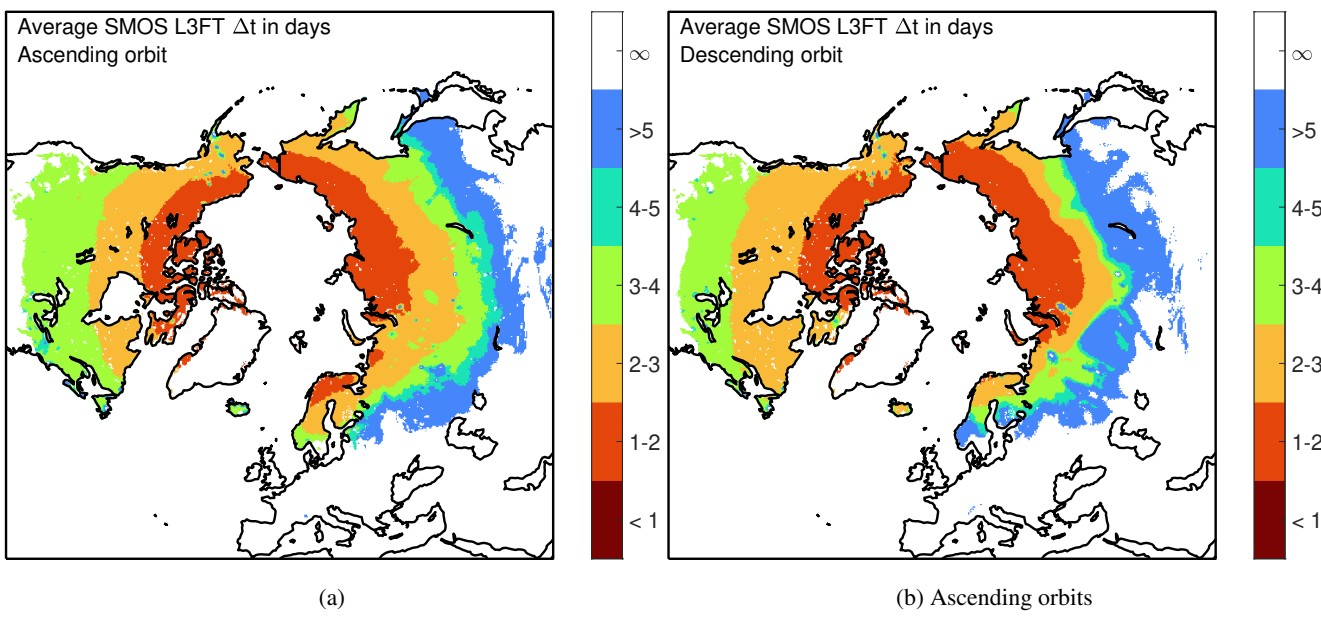

(a)

(b) Ascending orbits

**Figure 7.** Average observation interval of the SMOS FT product measured in days. The average is computed between 1 June 2010 and 31 Dec 2021 for (a) ascending and (b) descending orbits.

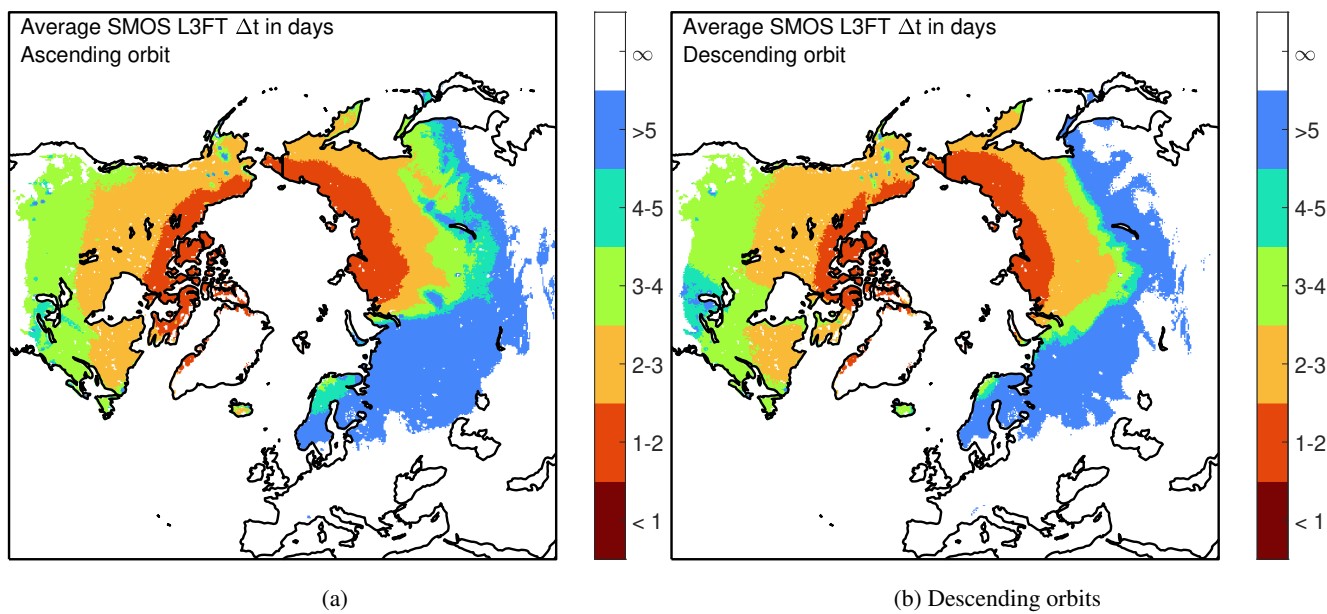

(a)

(b) Descending orbits

**Figure 8.** Same as Figure 7 but for the time period 1 Jan 2022 to 1 June 2024.



## 5.3 Wet snow

The presence of a wet snow hampers the ability of SMOS to detect soil conditions. In particular, this obscures the detection
of spring soil thawing. The signal from the surface soil layer beneath the wet snow is greatly attenuated and in the worst case
completely blocked. In addition, the wet snow layer itself causes a similar change in the observed brightness temperature as the
thawing soil, leading to misinterpretation of the observations. Variations in L band brightness temperature in spring should thus
rather be interpreted as information about the presence of liquid water in snow (Rautiainen and Holmberg, 2023). However, due
to partial penetration of L band microwave radiation even in wet snow, the interpretation of the signal is less straightforward
than at higher frequencies. On the other hand, this carries the potential to retrieve the liquid water content of snow (Houtz et al.,
2021) and also density of snow (Schwank et al., 2015; Lemmetyinen et al., 2016; Naderpour et al., 2017).

## 6   Conclusions

The SMOS FT product provides daily monitoring of the freeze-thaw (FT) state of Northern Hemisphere land surfaces at a
spatial resolution of 25 km. The first operational SMOS FT product, made public in 2018, was developed from the prototype
algorithm presented by Rautiainen et al. (2016). The updated SMOS FT product (version 3.01), presented here, offers a tool for
monitoring seasonal freeze-thaw cycles, particularly across high-latitude regions. The L band passive microwave observations
used in this product are effective in detecting soil FT transitions due to the sensitivity of L band brightness temperatures to
changes in soil permittivity between frozen and thawed states.

   The updated SMOS FT algorithm incorporates several improvements. These include enhanced noise removal through tempo-
ral averaging of the SMOS signal, which has improved the accuracy and reliability of the freeze-thaw detection. The validation
of the SMOS FT estimates against *in-situ* SM and ST data from international soil moisture network, along with comparisons
to the ERA5 Land reanalysis soil temperature data, demonstrate the product's robustness in identifying the day of the first
freezing in autumn, a critical parameter for greenhouse gas emissions studies.

   However, certain limitations do remain. The SMOS FT product is less effective in regions with dry soils, thin soil layers,
dense forested regions, or areas with significant radio frequency interference (RFI), particularly in Eurasia. Additionally, the
presence of wet snow in spring can obscure soil thawing detection, and variations in L band signals during spring should be
interpreted as an indication of wet snow rather than soil conditions; however, unambiguous detection of wet snow from L band
is itself also more challenging than at higher frequencies, due to partial penetration in wet snow. Furthermore, after the spring
of 2022, the exceptionally strong presence of RFI over Eastern Europe hinders the SMOS FT product on a large areas.

In conclusion, while the SMOS FT product shows strong performance in high-latitude environments, future work should
focus on addressing the limitations posed by RFI and wet snow layers. Continued refinement of the algorithm and further
validation in different environmental conditions will enhance the product's utility for climate change studies, ecosystem moni-
toring, and land-use management.

   Additionally, SMOS FT data have been utilized in the CarbonTracker Europe inverse modeling system at the Finnish Me-
teorological Institute to improve methane flux estimates at high latitudes. By aiding in the characterization of cold-season

emissions, the integration of SMOS FT data has demonstrated its value in reducing uncertainties and supporting studies of methane dynamics in northern ecosystems (Erkkilä et al., 2023; Tenkanen et al., 2021).

*Data availability.* The operational SMOS L3FT data record DOI is https://doi.org/10.57780/sm1-fbf89e0 (ESA, 2023). Data are available from ESA SMOS online dissemination service: https://doi.org/10.57780/sm1-fbf89e0 and from the FMI dissemination service: https://litdb.
fmi.fi/outgoing/SMOS-FTService/OperationalFT/. The associated documentation, the Algorithm Theoretical Baseline Document, Product Description Document and Read-me-first note are available from both services.

*Author contributions.* KR developed the original soil freeze and thaw detection algorithm and led the product updates for this study. KR and MH analyzed the product output and conducted its validations. MH was also responsible for the algorithm's modifications and implementation of new mathematical tools. JC contributed to algorithm scripting and GIS tool integration. JL and MS provided expertise in L band
remote sensing, particularly in snow-covered areas. AM advised on the use of CATDS input data. AdlF served as the ESA technical officer for the operational SMOS soil F/T product. YK led the SMOS Expert Support Laboratory project and offered valuable scientific advice related to SMOS data. KR and MH were the primary contributors to the writing of this article, with all other authors contributing to writing and revising the manuscript.

*Competing interests.* The contact author has declared that none of the authors has any competing interests


*Acknowledgements.* This work was funded by the European Space Agency (ESA - ESRIN), projects SMOS L3 Freeze/Thaw L3 Data Service (ESRIN Contract No: 4000124500/18/I-EF SMOS F/T Service) and ESA SMOS ESL2020+ (ESRIN Contract No: 4000130567/20/I-BG), the Research Council of Finland Academy Research Fellowship project EMonSoil (grant no. 364034), and the Research Council of Finland Flagship programs Advanced Mathematics for Sensing Imaging and Modelling (FAME; grant no. 359196). We acknowledge all the
data providers: The CATDS (Centre Aval de Traitements des Données SMOS) supported by CNES and IFREMER, for the polar gridded L3 brightness temperature data downloaded from https://data.catds.fr/cpdc/Common_products/GRIDDED/L3/ (CATDS, 2022). IMS Daily Northern Hemisphere Snow and Ice Analysis data were downloaded from https://noaadata.apps.nsidc.org/NOAA/G02156. International Soil Moisture Network (ISMN) data were downloaded from https://ismn.earth/en/dataviewer/. ESA CCI Land Cover data were downloaded from http://maps.elie.ucl.ac.be/CCI/viewer/download.php, and the ERA5-Land data were downloaded from https://cds.climate.copernicus.
eu/datasets/reanalysis-era5-land. This manuscript has been reviewed for language clarity using an AI-assisted tool (ChatGPT). The authors take full responsibility for the content and interpretations presented.



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
