# Peer review of "An operational SMOS soil freeze-thaw product"

_Earth System Science Data, 2025_

## Author Comment (AC1)

**Response to Referee Comments**

We sincerely thank the reviewers for their thorough and constructive evaluations of our manuscript. Their comments have helped us to improve the clarity, depth, and overall quality of the work. In this document, we provide detailed responses to each comment, indicating how the manuscript has been revised or, where applicable, providing clarification. Reviewer comments are shown in blue and italicized, followed by our responses in black.

**Response to Referee RC1**

**Referee RC1, comment 1.** *In the presentation and discussion of the derived DoFF results in Figure 5 the authors note differences between SMOS and ERA5 results at lower latitudes and in different regions such as Eurasia (e.g. Ln 295). However, these differences are difficult to distinguish in Fig. 5 as currently presented. It is recommended to add a SMOS-ERA5 DoFF difference map to this figure to more clearly show the regional difference pattern. Also, consider adding further discussion in this section regarding other factors contributing to the regional differences; e.g.: 1) impacts from potential greater NPR uncertainty over forests; 2) greater FT retrieval uncertainty in dry soil regions such as Tibetan Plateau; 3) greater DoFF differences over complex topography and in more intense RFI zones. Some of these issues are briefly mentioned as potential limitations in the Conclusions section (Ln 360), but more discussion should be given here in the results section, particularly as they may help explain the DoFF difference pattern.*

**Answer RC1, comment 1.** We have revised Figure 5 (now Figure 6) to include three additional panels that present DoFF difference maps:

- SMOS ascending minus ERA5,

- SMOS descending minus ERA5, and

- SMOS ascending minus descending.

These maps help highlight spatial differences in DoFF estimates and reveal regional patterns that were not apparent before.

Additionally, we have expanded the discussion in Section 4.2 to describe potential contributing factors in more detail. The second and third paragraphs of the section are now as follows:

*Figure 6 shows maps of the average DoFF derived from SMOS FT ascending and descending orbits separately, and from the ERA5 soil temperature product. SMOS FT and ERA5 show broadly similar DoFF patterns, particularly at high latitudes. However, discrepancies become more evident at lower latitudes, where SMOS has fewer observations and is more affected by RFI, especially in Eurasia. Differences also reflect expected latitudinal variation in DoFF dynamics.*

*To better highlight spatial differences, we introduce DoFF difference maps in Figure 6 (d–f). These show SMOS ascending minus ERA5 (d), SMOS descending minus ERA5 (e), and SMOS ascending minus descending (f). All maps use a centered colour scale to emphasize spatial variability. The SMOS FT product tends to estimate later freezing compared to ERA5, with median differences of +10.7 and +12.3 days for ascending and descending overpasses, respectively. These differences are spatially heterogeneous. Notably, larger DoFF differences occur in regions with dense forest cover, particularly in boreal Eurasia and parts of North America, where increased vegetation canopy attenuates the L-band signal and amplifies NPR uncertainty; in regions affected by strong RFI, such as Eastern Europe and parts of Russia, where the SMOS observation density is lower and retrieval quality is reduced; and in mountainous or topographically complex terrain (e.g., Scandinavia, Alaska), where sub-grid heterogeneity can lead to mismatches between model-based and radiometric observations.*

Regarding the Tibetan Plateau, we acknowledge the suggestion but note that this region is not covered in our SMOS FT product due to persistent RFI contamination and thus is not included in the DoFF comparison.

**Referee RC1, comment 2.** *Ln 190: How much of the classification domain and record is affected by the processing mask imposed frozen (PM: 5,6) and thaw (PM: 1,2) rules defined from the auxiliary air temperature and snow data (Table 3)?*

**Answer RC1, comment 2.**

To clarify the impact of the processing mask on the SMOS FT classification, we performed a dedicated analysis over a representative one-year period (June 1, 2016 to June 1, 2017) to quantify the number of observations flagged for correction based on the mask values PM = 1,2 (forced thaw) and PM = 5,6 (restricted thaw).

Figure 1 illustrates the frequency of such misclassifications prior to the application of processing mask constraints:

- Panel (a) shows the number of observations classified as frozen during the summer period (i.e., when PM = 1 or 2). These cases are forced to thaw in the SMOS L3FT processor (see Section 3.6 in the manuscript). Overall, such false detections are rare, with the exception of certain regions in North America. We suspect these cases relate to agricultural zones with dry surface conditions, or areas where the frozen soil reference is poorly defined due to short freezing periods.

- shows the number of observations classified as thaw during winter (PM = 5 or 6). These cases are more frequent, especially in Northern Siberia. In these cases, rising NPR during deep winter may result from snow densification or layering. Rain-on-snow events or mid-winter thaws also contribute, and we acknowledge that some of these cases may reflect real physical conditions (e.g., thawed soils under thick snow insulation).

Importantly, during winter, the L3FT processor applies a soft constraint: it prevents transitions into thawed states, but retains thawed classifications that persist from earlier periods.

We have not revised the manuscript text in this case, as the detailed mechanics of the processing mask are already described in Section 3.6 and Table 3, and we believe adding this level of statistical detail might distract from the algorithm's general description. However, we appreciate the opportunity to elaborate on this point here and provide this analysis for transparency and completeness.

[Figure]

(a)              (b)

Figure 1: Proportion of SMOS observations classified as (a) frozen during summer and (b) thaw during winter, prior to applying the processing mask correction (June 2016–June 2017). Values are expressed as a percentage of total observations during the period.

[Figure]

**Referee RC1, comment 3.** *Ln 202: Clarify how much of the domain and record is screened due to RFI?*

Figure 2: Maps showing (a) the average number of observations per day, and the average number of different quality flags raised per observation (b-f). The different quality flags are the ones mentioned in the Table 1 of the manuscript. SMOS observation is filtered out if at least one of the flags is raised during processing of the daily files. The results in this Figure are shown for ascending orbit and time period June 2017 - June 2018.

**Answer RC1, comment 3.** In the SMOS L3FT processor, five quality screening criteria are applied to the brightness temperature data (see Table 1 in the manuscript). Of these, four—namely the realistic brightness temperature threshold, RFI flag, chi-squared max threshold, and chi-squared min threshold—are directly or indirectly associated with RFI contamination. The remaining criterion (minimum number of angular views) is not inherently related to RFI.

To illustrate the relative impact of each criterion, Figure 2 shows the average number of quality flag rejections (per grid cell) for each criterion over the ascending orbit data between June 2017 and June 2018, normalized by the number of available observations (panel a). These results highlight that the RFI flag and chi-squared max threshold contribute most significantly to the spatial screening, particularly over Eurasia. This confirms that RFI remains a major factor limiting usable observations in the SMOS FT processor, especially in regions with frequent interference events.

While the other criteria also flag a small number of cases, their spatial patterns are more localized. The contribution from the minimum number of views and unrealistic brightness temperature checks is negligible over most areas. We note, however, that the role of the chi-squared max flag, especially in RFI-dense areas, requires further investigation to disentangle whether high chi-squared values stem from real signal uncertainty or residual RFI not caught by the primary RFI flag.

**Referee RC1, comment 4.** *Ln 224: Clarify range in SMOS local sampling times used to derive the FT retrievals; some of this detail is given later in the paper, but is also needed in this section. Also, be more specific regarding the latitude above which SMOS provides daily coverage.*

**Answer RC1, comment 4.** We have clarified in the manuscript that although combined SMOS overpasses provide daily coverage north of ∼60°N, our L3FT algorithm processes ascending and descending orbits separately, which means that near-daily coverage for either orbit is only achieved above ∼65°N. We now also refer to Figure 7 in the manuscript, which illustrates the spatial variability in SMOS observation frequency, and we have added a supporting reference to (Kerr et al. 2010).

We have modified the text as follows:

*Due to SMOS's orbit configuration, global coverage is achieved every three days, with combined ascending and descending overpasses enabling daily observations north of approximately 60°N (Kerr et al. 2010). However, because the SMOS L3FT retrievals are computed separately for each orbit, near-daily coverage for either ascending or descending observations is only attained at latitudes above ~65°N. This is also evident in the SMOS observation frequency map shown in Figure 8. Additionally, data quality filtering, especially due to radio-frequency interference (RFI), further reduces the effective observation frequency, particularly in Eurasia. As a result, the five observations required to confirm freezing typically span 5–15 days, depending on latitude and data quality. This limited temporal resolution may delay the DoFF detection relative to the actual onset of soil freezing.*

**Referee RC1, comment 5.** *Ln 291: "each data sets" should be "each data set".*

**Answer RC1, comment 5.** This is now corrected

**Referee RC1, comment 6.** *Ln 364: "on a large areas" should be "over large areas".*

**Answer RC1, comment 6.** This is now corrected

**Response to Referee RC2**

**Referee RC2, comment 1.** *Lines 112-119: This paragraph describes the behavior of L-band Tb as a function of the soil FT state, but it does so without reference to polarization. The SMOS FT algorithm, however, is based on the Normalized Polarization Ratio (NPR; eq. 1). That is, the physical basis discussed here is not sufficient to motivate why and how NPR can be used to derive the FT status. If both H- and V-polarization Tb are changing identically with soil freezing and thawing, the NPR would not provide any information about the soil FT status. Please explain how polarization impacts soil FT and relate the discussion to NPR as the basis of the FT algorithm.*

**Answer RC2, comment 1.** We have now substantially revised Section 3.1 (Algorithm outline) to clarify the physical rationale for using the Normalized Polarization Ratio (NPR). The revised section now explains how soil freeze–thaw transitions affect both horizontal and vertical emissivities—both in absolute terms and in polarization contrast. A new figure (Figure 3, below) was added to illustrate the modeled NPR response to varying soil permittivity at the SMOS incidence angle bin used in FT algorithm.

[Figure]

Figure 3: Permittivity

**Referee RC2, comment 2.** *The same comment applies to Lines 312: "minimal dynamics of the Tb signal" I see that that it's difficult to estimate anything from a constant signal, but NPR is different from the "signal" discussed here. The basis of the FT algorithm is NPR, not DeltaTb. The same comment also applies to Lines 341-342.*

**Answer RC2, comment 2.** We agree that the original text referred imprecisely to the brightness temperature signal, whereas the SMOS FT algorithm is based on the Normalized Polarization Ratio (NPR), not directly on brightness temperature $T_{\mathrm{B}}$ or its temporal variation.

In Section 5.1, we revised the text to clarify that in dry soil conditions, the dielectric contrast between frozen and thawed states is inherently small, which leads to low variability in polarization contrast and therefore limited sensitivity in NPR. This reduced NPR dynamic range, rather than brightness temperature dynamics per se, makes freeze–thaw transitions more difficult to detect in such regions.

In Section 5.3, we revised the wet snow discussion to accurately describe how wet snow increases polarization contrast, resulting in high NPR values that resemble those of thawed, moist soil. While snow melt and soil thaw typically occur at the same time, the L-band signal cannot unambiguously distinguish between these processes. One new citation was added to the text related to the wet snow observations at L-band (Pellarin et al. 2016)

These clarifications align the text more closely with the physical basis of the algorithm and address the reviewer's concern regarding appropriate terminology and interpretation.

The section 5.1 now starts as follows:

*The SMOS FT retrieval algorithm detects permittivity changes caused by the phase transition (or change in the aggregate state) of liquid soil water to ice. In regions with dry soils, this permittivity change is inherently small because the soil already has a low dielectric constant—even when unfrozen. As a result, the polarization contrast, and consequently the variability in the Normalized Polarization Ratio (NPR), is limited. This low NPR dynamics reduces the sensitivity of the algorithm to freeze–thaw transitions in such environments.*

The section 5.3 starts now as follows:

*The presence of wet snow hampers the ability of SMOS L-band observations to detect soil conditions, particularly during spring. As snow begins to melt, its high liquid water content attenuates the microwave signal from the underlying soil and strongly affects the observed brightness temperatures—especially in horizontal polarization (Pellarin et al., 2016). This leads to increased polarization contrast and elevated values of the Normalized Polarization Ratio (NPR), similar to those observed for thawed, moist soil. While snow melt and soil thaw often occur concurrently in spring, the timing and depth of thaw can vary, and the L-band signal cannot unambiguously distinguish between wet snow and actual thawed soil. Consequently, the SMOS FT algorithm may misinterpret the presence of wet snow as an early soil thaw, introducing uncertainty in the retrieval during the spring melt period.*

**Referee RC2, comment 3.** *Line 75: The T2m data are introduced as being from ECMWF's operational "High-Resolution 10-day Forecast", but the information about the resolution of these forecasts isn't provided until Line 80, after the introduction of the ERA5-Land reanalysis. I assume the "High Resolution" is the 0.1-deg-by-0.1-deg mentioned in Line 80, but it's not entirely clear whether this resolution refers to the operational "High-Resolution" forecast or to ERA5-Land or both. Please clarify the resolution of the ops forecasts and of ERA5-Land.*

**Answer RC2, comment 3.** Text is now modified accordingly:

*Both the operational 10-day high-resolution forecasts and the ERA5-Land reanalysis from ECMWF are provided on a grid with a spatial resolution of $0.1° \times 0.1°$ (approximately 11.1 km $\times$ 11.1 km at the equator, and 11.1 km $\times$ 5.6 km at 60° latitude) ...*

**Referee RC2, comment 4.** *Line 77: The temperature from ERA5-Land is referred to as "surface layer" data. What exactly do you mean by that? Is it also T2m (as the section heading implies), or is it the temperature of the lowest model layer of the atmospheric model that provides the ERA5-Land surface met forcing, or is it a "surface" or "skin" temperature? Note that section 2.2.2 mentions T2m from ERA5-Land, but in the context of validation of the FT product. It's not entirely clear which air*

*temperature from ERA5-Land was used in the retrospective processing of the SMOS FT estimates. Please clarify.*

**Answer RC2, comment 4.** This is now corrected, the sentence is now:

*During reprocessing, the near real-time air temperature data are replaced with the corresponding air temperature data from the ERA5-Land reanalysis, which are available with a latency of up to one month ...*

**Referee RC2, comment 5.** *Line 85: My understanding is that IMS is primarily based on optical data, which cannot be used for snow cover detection when clouds are present. But the IMS product is referenced as a "Daily" product. I suppose some temporal interpolation or persistence is used in deriving the "Daily" IMS product, with obvious implications for the quality of the IMS estimates on days with cloud cover. This limitation of the IMS observations should be mentioned here.*

**Answer RC2, comment 5.** IMS is a composite of optical and microwave data, when optical data is not available (clouds, high latitudes during winter), the product relies on passive microwave data and the resolution is degraded. For the SMOS purposes this is sufficient, though. We have added sentences at the end of section 2.1.3 Snow extent accordingly:

*Although IMS provides daily global snow extent, its quality may be affected by persistent cloud cover and polar night conditions, which limit the availability of optical observations. In such cases, the IMS algorithm relies more on passive microwave inputs and temporal persistence from previous days' estimates (U.S. National Ice Center, 2008, updated daily; S. Helfrich, M. Li, C. Kongoli, L. Nagdimunov and E. Rodriguez, 2019). Given the coarse spatial resolution of the SMOS L3FT product, these limitations are not considered critical for our application.*

**Referee RC2, comment 6.** *Line 103: "using [...] nearest-neighbor interpolation". Since the ERA5-Land data are on a 0.1-deg grid, why are you not aggregating the ERA5-Land data to the 25km resolution of the SMOS FT data? Or is this validation somehow related to point locations (of, say, the in situ measurements discussed in section 2.2.1)?*

**Answer RC2, comment 6.** We are indeed using averaging, not nearest-neighbor interpolation, when resampling the ERA5-Land data to the 25 km resolution of the SMOS L3FT product. This is achieved using the GDAL utility `gdalwarp` in two steps: first, the data are reprojected from geographic coordinates (EPSG:4326) to the EASE2 North projection (EPSG:6931); second, they are resampled to the target resolution using average aggregation. The relevant commands in our implementation are:

- First command:

  ```
  gdalwarp -s_srs EPSG:4326 -t_srs EPSG:6931 -srcnodata 0 -dstnodata 0
      inputfile outputfile
  ```

- Second command:

  ```
  gdalwarp -te -9000000 -9000000 9000000 9000000 -tr 25000 25000 -r average -
      srcnodata 0 -dstnodata 0 inputfile outputfile
  ```

We have now clarified the manuscript text accordingly. The updated sentence reads:

*The data, provided on a 0.1° × 0.1° latitude–longitude grid, are reprojected to the 25 km × 25 km EASE-2 grid used by the SMOS L3FT product. This is done using the Geospatial Data Abstraction Library (GDAL), with average resampling applied during resolution matching to ensure consistency with the SMOS grid.*

**Referee RC2, comment 7.** *Line 105: What is the spatial resolution of the ESA CCI Land Cover data?*

**Answer RC2, comment 7.** The original spatial resolution is 300 m, we aggregate it to the 25 km EASE-2 grid. We have modified the paragraph as follows (new text in bold font):

The ESA CCI Land Cover time series v2.0.7 (1992 - 2015) data (Muñoz Sabater et al., 2021), **originally provided at 300 m spatial resolution**, are used to define the land cover distribution on the EASE-2 grid. The land cover classes were aggregated from the original 23 classes into 6 classes: agriculture, forest, low vegetation, wetland, open water and other (permanent ice, barren, urban). This aggregated land cover information was then **regridded to the 25 km EASE-2 grid and** used during the validation process to determine whether the land cover class at each in-situ sensor location represented the larger EASE-2 grid cell.

**Referee RC2, comment 8.** *Line 138: "bounded both from above and below with values 2 and 0.1, resp". The need for an upper bound is obvious. But why should there be a lower bound? Please explain your motivation for choosing a lower bound on the noise estimate of (eq. 2).*

**Answer RC2, comment 8.** We are loosely following a chi-squared test for normal distributions, where the following quantity in our manuscript

$$\chi^2 = \left( \frac{\text{STD}}{\text{ACC}} \right)^2 \tag{1}$$

is thought to be distributed according to chi-squared distribution (as it is empirical variance divided by the theoretical variance). Consequently, $\chi^2$ should follow chi-squared distribution (normalized with the number of views), and:

- Probability that this quantity obtains a high value (i.e. that there is considerably more spread in the brightness temperature values inside the incident angle bin that what is predicted by the radiometric accuracy) is low. This corresponds to the upper bound $\chi \leq 2$ used in the SMOS L3FT algorithm. If a high value is obtained, we suspect RFI contamination.

- Similarly, the probability the the spread is considerably lower than what is predicted by the radiometric accuracy value is low as well. This corresponds to the lower bound $0.1 \leq \chi$ used in the SMOS L3FT algorithm. If a low value is obtained, we suspect other issues with the data, or uncertain data (high value for radiometric accuracy variable, i.e. low actual accuracy).

Our approach deviates from the statistical chi-squared test in the choice of the thresholding values 0.1 and 2, which were determined empirically based on visual inspection of the filtered observations. Also, in theory the thresholds should depend on the number of views within the incident angle bin. However, again, we obtained better results by empirical thresholds. A possible reason for these discrepancies to the theoretical values is that the radiometric accuracy is systemically under estimated in the SMOS L3TB product. Also it is important to note that we are only considering instances where the number of observations within the incident angle is $\geq 5$.

See also our answer to Referee RC1, comment 3. Figure 2b shows the number of flags raised due to the lower bound $0.1 \leq \chi$. The amount of flagged observations is very low and on areas where other flags are raised, too.

**Referee RC2, comment 9.** *Line 139: "the proportion of measurements suspected to be contaminated by RFI within the incident angle bin must be less than 40%." From this I understand that the FT estimate is set to no-data if $N\_RFI/N\_views > 40\%$. But what if $N\_RFI/N\_views <= 40\%$? Are the RFI-impacted angular data included in the computation of the bin average that is used to derive the FT estimate? I would assume that only the "good" data are included, but I didn't see an explicit statement to that effect. And if the RFI-impacted data are excluded, is the $N\_views >= 5$ threshold applied before or after excluding the RFI-impacted data?*

**Answer RC2, comment 9.** Our SMOS L3FT algorithm uses brightness temperatures from the CATDS L3TB data set. The CATDS L3TB data set contains already incidence-angle-bin-averaged brigthness temperature values. Therefore, unfortunately the individual brigthness temperature values cannot be separated from the averages anymore at this stage, and we have to work with compromise of filtering out RFI contaminated observations while keeping as much clean data as possible.

The RFI flag provided in the L3TB data expresses number of *suspected* RFI instances. The ratio of suspected RFI-contaminated observations to the total number of views in the bin (i.e., $N_{\text{RFI}}/N_{\text{views}}$)

is used as an empirical indicator of data quality. If this ratio exceeds 40%, we discard the brightness temperature for that bin. This 40% threshold was selected based on empirical analysis: it provides a reasonable balance between retaining sufficient data coverage and avoiding significant RFI bias.

Regarding the $N_{\text{views}} \geq 5$ threshold: it is applied in parallel with the RFI check. That is, both conditions (a minimum number of observations and an acceptable RFI contamination ratio) must be satisfied for the brightness temperature to be used. Since both checks operate on the same L3TB summary information, their order of application does not affect the outcome.

After this filtering step, the NPR values are computed from the valid L3TB brightness temperatures. If no valid data are available for a given day and location (due to insufficient views or high RFI), a no-data value is assigned to NPR. When the Kalman filter is applied, these no-data instances are handled by propagating the previous day's estimate forward. While this allows continuity in the time series, we also provide metadata in the output indicating the time since the last successful observation, enabling users to mask or filter uncertain retrievals if desired.

We added clarification to the beginning of the section 3.2 as follows:

*The SMOS L3FT processor uses CATDS L3TB data, which are already averaged within incidence angle bins; hence, the quality flags, including suspected RFI proportion, are interpreted as summary statistics, and individual brightness temperature measurements are no longer accessible at this stage.*

**Referee RC2, comment 10.** *Line 144: replace "independently from each other" with "independently from every other"?*

**Answer RC2, comment 10.** This is corrected in the text

**Referee RC2, comment 11.** *Line 179: "The threshold have been acquired...". Clarify as follows: "The thresholds of 50*

**Answer RC2, comment 11.** This is now clarified and corrected.

**Referee RC2, comment 12.** *Line 185: For a given season, does the processing mask mentioned here vary from year to year or is it a seasonally varying climatological mask? Please clarify.*

**Answer RC2, comment 12.** The processing mask is not a fixed climatological mask but is computed dynamically for each grid cell and day based on auxiliary data. Specifically, it uses the 2-metre air temperature (from ECMWF forecast or ERA5-Land reanalysis) and daily IMS snow extent. Additionally, the state of the processing mask from the previous day is used to constrain allowable transitions, following the transition rules defined in Table 3 of the manuscript. This ensures temporal consistency in the mask and helps prevent unrealistic fluctuations. The primary function of the processing mask is to suppress false FT detections during summer when dry soils or other geophysical factors may mimic freeze signatures in the brightness temperature signal.

**Referee RC2, comment 13.** *Line 192: "However, the frozen state is not forced." Does this include the "frozen and partially frozen state"? Please clarify.*

**Answer RC2, comment 13.** Yes, this includes both frozen and partially frozen states. We have clarified the text as follows:
*However, neither the frozen state nor the partially frozen state is forced.*

**Referee RC2, comment 14.** *Line 195: "The spatial and temporal differences between..." I assume by "spatial .. differences" you mean the "scale mismatch" (point scale vs. grid cell scale). Perhaps clarify further. What do you mean by "temporal differences"? In situ measurements are usually available hourly, so sub-selecting the in situ measurements to the times of the SMOS overpasses should address the "temporal differences" (as discussed in Lines 200-201). The fact that RFI impacts the sampling of SMOS simply reduces the number of available Tb observations (Lines 202-203), but the "good" Tb observations*

*and available FT retrievals can still be time-matched with the hourly in situ measurements. So it's not clear to me what you mean by "temporal differences" in Line 195. Please clarify.*

**Answer RC2, comment 14.** We agree that "spatial differences" is more accurately described as scale mismatch between point-based *in-situ* measurements and the large satellite footprint. Regarding "temporal differences," we did not mean time-matching issues per se, but rather the uncertainty in the timing of freeze events in SMOS data due to non-daily observations and frequent RFI-induced gaps. Even though in-situ data can be subsampled to match SMOS overpass times, the sporadic nature of available SMOS observations means the satellite may not observe the true onset of freezing in a timely manner. We have revised the manuscript to clarify both of these points accordingly. We have revised the first paragraph of this section as follows:

*The soil freeze–thaw (FT) estimates were compared against the ISMN SM and ST data. The scale mismatch between satellite-based and in-situ observations presents significant challenges when interpreting the comparison results. In-situ sensors measure the soil state at a single point, whereas SMOS observations represent an area with an effective footprint of 30–50 km, depending on location within the snapshot scene (McMullan et al., 2008; Kerr et al., 2010).*

*Temporal uncertainty also affects the comparison because SMOS does not always provide daily observations for a given location. Due to its orbital configuration and data gaps caused by radio frequency interference (RFI), particularly in Eurasia (Olivia et al., 2016), the satellite may miss critical transition days. This can delay or obscure the detection of the actual freeze onset. In contrast, in-situ data are typically available at hourly resolution, allowing precise identification of freezing events. Although SMOS and in-situ data can be time-matched when observations are available, the discontinuous temporal sampling of SMOS introduces uncertainty that must be considered in the comparison.*

**Referee RC2, comment 15.** *Lines 220-221: "in situ sensors... are not the most suitable ground reference for validating SMOS results during spring". This seems a bit backwards to me. If SMOS FT estimates are indicative of wet snow rather than soil FT, then the issue is more that SMOS FT estimates do not reflect the soil FT state, and the product name or objective is not suitable. There's nothing wrong with the in situ measurements. Please rephrase.*

**Answer RC2, comment 15.** We agree with this comment. Our intention was not to discredit the *in-situ* data, but rather to highlight that during the spring melt, the SMOS FT retrieval becomes insensitive to the actual soil state due to strong attenuation by wet snow. We have revised the sentence to better reflect this limitation of the SMOS product, and clarified that the issue lies in the satellite's ability to detect soil conditions during spring—not in the suitability of the *in-situ* measurements. We have revised the sentences as follows:

*As a result, the SMOS FT estimates during this period often reflect the condition of the snowpack (e.g., presence of wet snow) rather than the actual soil thawing. While in-situ sensors provide accurate information about the soil state itself, the springtime SMOS FT signal cannot be directly interpreted as soil thaw, which limits its suitability for soil FT validation during this season.*

**Referee RC2, comment 16.** *Line 231: "we identified the first time after which the soil state potentially changed to frozen". Does this refer to the first frozen FT status data \*without\* applying the 5 consecutive frozen obs requirement? Please clarify.*

**Answer RC2, comment 16.** The original phrasing was indeed unclear and may have suggested that the five consecutive frozen observations applied to both the day of first freezing (DoFF) and the day of first potential freezing (DoFPF). However, this is not the case.

We have clarified the manuscript text to explain that DoFPF refers to the last thawed observation prior to the onset of freezing, and does not require five consecutive frozen observations. It is used to define the start of the uncertainty window within which the freezing transition occurred. In contrast, DoFF is defined as the first day of five consecutive frozen state observations, confirming the onset of persistent freezing. This distinction now reads as follows in the revised text:

*To account for temporal uncertainty due to irregular SMOS sampling, we define the day of first potential freezing (DoFPF) as the last observation that still indicated a thawed state before the confirmed onset of freezing (DoFF). This ensures that the actual transition lies between DoFPF and DoFF. The period between these two dates represents the time during which SMOS FT estimates indicate the onset of soil freezing in autumn.*

**Referee RC2, comment 17.** *Line 244: "with in situ measurements providing clearer signals of soil state changes". In the graphic, the horizontal error bars appear to be no shorter than the vertical error bars. What do you mean by "clearer signals"?*

**Answer RC2, comment 17.** We acknowledge that the phrasing "clearer signals" could be misleading when interpreted solely from the graphic. Our intent was not to imply that the error bars from *in-situ* measurements are necessarily shorter, but rather that the temporal resolution and physical proximity of the sensors to the soil allow a more precise and immediate detection of freeze–thaw transitions when the signal is unambiguous.

We have revised the sentences to better reflect this intended meaning:

*A wider range between the 50% and 90% thresholds suggests more gradual soil freezing, introducing greater uncertainty into the timing of DoFF. In contrast, narrower error bars indicate a more abrupt freeze transition and therefore a more certain timing estimate at the sensor location.*

**Referee RC2, comment 18.** *Tables 4 and 5: Are the numbers of data points (N) valid for separate or cumulative application of the checks (a)-(c)? That is, does N=133 for "SFD check (c)" reflect the application of just "SFD check" or the simultaneous application of "LC check", "FDD check" and "SFD check"? Please clarify.*

**Answer RC2, comment 18.** The representativeness checks are applied cumulatively: the "FDD check (b)" includes only those data points that passed the "LC check (a)", and the "SFD check (c)" includes only those that passed both the LC and FDD checks. We have clarified this in the captions of Tables 4 and 5. Specifically, Table 4 now includes a sentence explaining the cumulative nature of the checks with an example, while Table 5 includes a shorter note confirming the same cumulative approach.

**Referee RC2, comment 19.** *Line 275: "the bias is -6.3 days...". Why are the numbers in the text different from those in Table 5? Line 281: "the bias is -5.0 days...". Why are the numbers in the text different from those in Table 4?*

**Answer RC2, comment 19.** The differences arise because the text in lines 275 and 281 refers to the midpoint values of the error bars shown in Figure 8, which are computed as the average between DoFPF and DoFF for SMOS, and between the 50% and 90% SFTI thresholds for the in-situ measurements. This midpoint better reflects the central estimate when uncertainty is taken into account.

In contrast, the bias values reported in Tables 4 and 5 are derived from a direct comparison between the SMOS DoFF estimates and the in-situ DoFF determined using only the 50% SFTI threshold, without incorporating the range or midpoint of the uncertainty bars.

We have added a clarification in the text as follows:

*Note that these metrics are based on the midpoint values of the uncertainty ranges shown in the figure: for SMOS, the midpoint between the day of first potential freezing (DoFPF) and the day of first freezing (DoFF), and for the in-situ data, the midpoint between the 50% and 90% SFTI thresholds. This differs from the comparison metrics in Tables 4 and 5, which are computed directly between SMOS DoFF and the in-situ DoFF derived from the 50% SFTI threshold.*

**Referee RC2, comment 20.** *Figure 4: The orange and red colors are difficult to discern. Likewise for the medium and dark blue colors. I suggest using a different marker for each entry in the legend. (Is there a reason why SNOTEL, USCRN, and FMI have the same marker but one that differs from the rest of the datasets?)*

**Answer RC2, comment 20.** We have revised the figure to improve clarity by assigning a distinct marker style to each dataset and adjusting the color palette to enhance visual contrast, especially between similar hues. These changes ensure that all datasets are now more easily distinguishable, particularly in the presence of overlapping error bars. We believe this significantly improves the interpretability of the figure.

[Figure]

(a) Ascending orbits          (b) Descending orbits

Figure 4: Revised Figure

**Referee RC2, comment 21.** *Line 302-303: "The estimation of the day of the first freezing from the two data sets is slightly different,..." It's not clear to me if the two data sets have been cross-masked. Please clarify.*

**Answer RC2, comment 21.** We understand the reviewer's reference to "cross-masking" to mean that the comparison between the SMOS and ERA5 DoFF estimates should be limited to those spatiotemporal locations where both datasets provide valid values. This is indeed the approach we followed: ERA5-derived DoFF values were computed and included in the analysis only for those grid cells and years where a corresponding SMOS FT estimate was also available.

The quoted sentence — "The estimation of the day of the first freezing from the two data sets is slightly different..." — refers to methodological differences in how DoFF is determined from each dataset. SMOS provides observations at irregular time intervals due to orbit configuration and quality filtering (e.g., RFI screening), whereas ERA5 offers regular daily data. This difference in temporal sampling introduces systematic uncertainty in the timing of DoFF from SMOS compared to ERA5.

**Referee RC2, comment 22.** *Figure 5: It's difficult to see the differences between the three panels. I suggest replacing panels (a) and (b) with difference maps (e.g., SMOS FT ascending minus ERA5)*

**Answer RC2, comment 22.** A similar point was raised by Referee RC1 (Comment 1), and we have addressed it by updating Figure 5 to include dedicated difference maps in panels (d–f). These maps illustrate the differences between SMOS FT (ascending and descending orbits) and ERA5 DoFF, as well as between the SMOS orbits themselves. The original panels (a–c) showing absolute DoFF values have been retained to provide full context. The new difference maps follow a centered color scale to emphasize spatial variability in the deviations, and this update has greatly improved the interpretability of the comparison.

**Referee RC2, comment 23.** *Figure 5: The validation period here is 2010-2024, but the reference values were derived after excluding 2010-2013 (Lines 173-174). Would the validation results change if 2010-2013 were excluded from the validation period?*

**Answer RC2, comment 23.** The validation results would not change significantly if the period 2010–2013 were excluded. While the reference values for threshold-based FT classification were indeed derived using data from 2014 onwards (mainly due to larger RFI contamination during the early SMOS mission years), the DoFF estimation and validation over the full period of 2010–2024 remain robust. Including 2010–2013 in the spatial validation ensures a longer time series and broader sampling, and any potential differences introduced by the early years have negligible impact on the aggregated DoFF metrics and spatial patterns shown in Figure 5.

**Referee RC2, comment 24.** *Figure 6: Plot 1:1 line (perhaps a thin light gray line will work ok)*

**Answer RC2, comment 24.** We have added a 1:1 line to these plots.

**Referee RC2, comment 25.** *Line 330: Arguably, RFI is a major problem with SMOS observations. But SMAP knew about the RFI environment before launch and includes tools to reduce the adverse impact of RFI. Please point to SMAP Tb observations as an alternative source of L-band passive microwave FT information that is somewhat less impacted by RFI.*

**Answer RC2, comment 25.** We have added a short text to the manuscript as follows:

*While SMOS has provided valuable L-band observations since 2010, RFI, particularly over parts of Eurasia, remains a significant challenge for data continuity. The Soil Moisture Active Passive (SMAP) mission, launched in 2015, also operates in the L-band and incorporates onboard RFI detection and mitigation techniques that help reduce the impact of radio interference in some regions (Piepmeier et al., 2014). As such, SMAP can serve as a complementary source of L-band brightness temperature data for freeze–thaw applications, particularly in areas where SMOS data quality is frequently compromised.*

**Referee RC2, comment 26.** *Line 355: replace "averaging" with "filtering" ? The KF isn't simply "averaging"*

**Answer RC2, comment 26.** This has been fixed in the paper.